# Effects of Type 2 Diabetes Mellitus on Osteoclast Differentiation, Activity, and Cortical Bone Formation in POSTmenopausal MRONJ Patients

**DOI:** 10.3390/jcm11092377

**Published:** 2022-04-23

**Authors:** Sung-Min Park, Jae-Hoon Lee

**Affiliations:** Department of Oral and Maxillofacial Surgery, College of Dentistry, Dankook University, 119 Dandae-ro, Cheonan 31116, Korea; 12201114@dankook.ac.kr

**Keywords:** type 2 diabetes, osteoporosis, bisphosphonate, MRONJ, osteoclast

## Abstract

Osteoporosis is a common metabolic bone disease in patients with diabetes, which can develop simultaneously with type 2 diabetes (T2D) in postmenopausal women. Bisphosphonate (BP) is administered to patients with both conditions and may cause medication-related osteonecrosis of the jaw (MRONJ). It affects the differentiation and function of osteoclasts as well as the thickness of the cortical bone through bone mineralization. Therefore, this study aimed to investigate the effects of T2D on osteoclast differentiation and activity as well as cortical bone formation in postmenopausal patients with MRONJ. Tissue samples were collected from 10 patients diagnosed with T2D and stage III MRONJ in the experimental group and from 10 patients without T2D in the control group. A histological examination was conducted, and the expression of dendritic cell-specific transmembrane protein (DC-STAMP) and tartrate-resistant acid phosphatase (TRAP) was assessed. Cortical bone formation was analyzed using CBCT images. The number of TRAP-positive osteoclasts and DC-STAMP-positive mononuclear cells was significantly less in the experimental group (*p* < 0.05). Furthermore, the thickness and ratio of cortical bone were significantly greater in the experimental group (*p* < 0.05). In conclusion, T2D decreased the differentiation and function of osteoclasts and increased cortical bone formation in postmenopausal patients with MRONJ.

## 1. Introduction

Diabetes is a metabolic disease with high morbidity that may be accompanied by various skeletal disorders such as osteoporosis, osteopenia, Charcot arthropathy, and diabetic foot disease [1]. Among these skeletal disorders, osteoporosis is the most important metabolic bone disease in diabetes patients, who are more likely to develop osteoporosis and fractures [2,3,4,5,6]. In particular, type 2 diabetes (T2D) and osteoporosis may manifest simultaneously in elderly postmenopausal women, and the risk of fractures increases in these patients [7,8,9,10]. T2D can directly affect bone metabolism and strength, and certain oral hypoglycemic agents may affect bone metabolism in T2D patients [11]. Additionally, unlike in type 1 diabetes, where osteoporosis or fractures are caused by decreased bone density, T2D is associated with increased bone density and the risk of fractures. Therefore, the standards for general osteoporosis are not satisfied in T2D patients [12].

Although previous findings show that osteoporosis in T2D patients increases bone fragility, its pathophysiology in T2D is different from that of conventional osteoporosis [5,7,8,9,10,11,12]. Therefore, studies are being actively conducted to seek the ideal treatment strategy [13]. Complex treatments, including weight and diet control, the use of antiresorptive and bone-forming agents, and the intake of calcium as well as vitamin D, are recommended; however, the resulting skeletal effect of such treatments is currently unknown [14].

In summary, most patients with both T2D and osteoporosis are treated in the same way as those without diabetes, and the most commonly used agent is bisphosphonate (BP) [15]. Therefore, patients with both T2D and osteoporosis may also suffer from BP-induced MRONJ.

BP affects the differentiation and bone resorption of osteoclasts [16,17,18,19]. When osteoclasts differentiate and function, biomarkers such as the dendritic cell-specific transmembrane protein (DC-STAMP) and tartrate-resistant acid phosphatase (TRAP) are expressed [20,21]. DC-STAMP is mostly expressed during the differentiation of osteoclasts and plays a key role in resorption and the cell–cell fusion between osteoclasts. It is a fundamental factor in the maturation of osteoclasts [22]. Mice deficient in DC-STAMP are unable to form multi-nucleated osteoclasts [23]. TRAP is a metalloenzyme and a known histochemical marker for osteoclasts [24]. Its secretion by osteoclasts leads to bone resorption [24], and in animals deficient in TRAP, osteoclast function is impaired, leading to increased bone density [25]. 

If the function of osteoclasts is impaired, the secondary mineralization of bone may occur, and subsequently, bone matrix density may increase. This may be observed as an increment in the thickness of the inner surface of the cortical bone on CT images, which results in increased cortical bone thickness and reduced cancellous bone area [26]. Increased cortical bone thickness is known to be associated with MRONJ [27], and we previously demonstrated that patients taking BP who developed MRONJ had a significant increase in cortical bone thickness and ratio compared with those who did not develop MRONJ [28]. Therefore, DC-STAMP and TRAP may play an essential role in analyzing the differentiation and activity of osteoclasts in bone samples of MRONJ patients and in inferring that the abnormal differentiation and activity of osteoclasts may affect cortical bone formation. 

The purpose of this study was to compare and analyze the expression of DC-STAMP and TRAP in bone samples from postmenopausal MRONJ patients with or without T2D and to investigate the thickness and ratio of the cortical bone near the mandibular mental foramen using CBCT. Furthermore, this study aimed to determine the effect of T2D on the differentiation and activity of osteoclasts and on cortical bone formation in patients with MRONJ who took BP for the treatment of osteoporosis.

## 2. Materials and Methods

### 2.1. Patient Selection and Specimen Harvesting

Retrospective analyses were performed for routine jawbone specimens obtained from 20 patients who underwent surgery for the treatment of clinically and histologically confirmed MRONJ at Dankook University Hospital (Department of Oral and Maxillofacial Surgery, Cheonan, South Korea) from February 2015 to February 2020. The 20 patients were divided into two groups according to whether the patients had or did not have T2D: (1) the experimental group included 10 patients who were diagnosed with MRONJ and T2D (treated with hypoglycemic agents) and were administered BP (orally) for the treatment of osteoporosis; (2) the control group included 10 patients who had not been diagnosed with T2D and were orally administered BP for the treatment of osteoporosis. To exclude potential cases of T2D in the control group, only patients with a fasting blood glucose level of less than 100 mg/dL in pre-operative tests were included. In both groups, no patients had systemic diseases or were on medications that could affect bone homeostasis, and all participants were postmenopausal women. All patients had stage III MRONJ (according to the MRONJ staging system previously described by the American Association of Oral and Maxillofacial Surgeons in 2014 [29]) and were taking oral alendronate (Fosamax™). The mean age and duration of BP administration were not significantly different between the two groups (Table 1).

During surgery, a sample from the jawbone of the patients was collected for histopathological examination. The collected tissues were fixed in 4% formalin, embedded in paraffin blocks, and stored at Dankook University Hospital, Department of Pathology. The paraffin blocks were obtained after approval from the Institutional Review Board (IRB) of Dankook University Dental Hospital (IRB number: DKUDH IRB 2019-07-003). 

The paraffin blocks were sliced into 6-µm-thick sections using a microtome (Leica Microsystems, Wetzlar, Germany), and each block was made into three sections.

### 2.2. Histochemistry 

#### 2.2.1. Hematoxylin and Eosin (H&E) Staining

A total of 20 sections from the experimental (*n* = 10) and control (*n* = 10) groups were deparaffinized and stained with H&E. The stained tissue samples were observed using an optical microscope (BX-41, Olympus Optical, Tokyo, Japan) and photographed with a Panoramic 250 Flash III scanner (3DHISTECH Kft, Budapest, Hungary) for digitization. As each sample had different sizes, two regions of interest (ROI, 5 × 5 mm) were set for each sample (Figure 1). Quantitative and morphological analysis of osteoclasts were conducted within the ROI. For quantitative analysis, the total number of osteoclasts in the ROI was assessed using Image-Pro Plus (Media Cybernetics, Rockville, MD, USA), and morphological analysis was conducted by investigating the diameter of the osteoclasts and the number of nuclei per osteoclast using a panoramic viewer. The mean and standard deviation (SD) were calculated. 

#### 2.2.2. Tartrate-Resistant Acid Phosphatase (TRAP) Staining

A total of 20 sections from the experimental (*n* = 10) and control (*n* = 10) groups were deparaffinized and stained for TRAP using TRAP detection system (TRACP & ALP double-stain Kit, MK 300, Takara Bio, Kusatsu, Japan). The images were digitized using an optical microscope and a scanner. Multi-nucleated cells with three or more nuclei with positive response to TRAP were considered to be osteoclasts, and the number of TRAP-positive osteoclasts per ROI was analyzed using Image-Pro Plus. The mean and standard deviation were calculated. 

### 2.3. Immunohistochemistry

In total, 20 sections from the experimental (*n* = 10) and control (*n* = 10) groups were deparaffinized. Antigen retrieval was performed for staining using ethylenediaminetetraacetic acid (EDTA). 

The tissues were cultured with anti-DC-STAMP antibody (Anti-DC-STAMP, rabbit polyclonal Atlas antibodies, Stockholm, Sweden) to detect the target protein. The antibody-marked proteins were visualized by treating the tissues with dextran and diaminobenzidine (DAB) chromogen. To detect the antibody, Zymed SuperPicTure polymer detection kit (Zymed, Invitrogen, Carlsbad, CA, USA) was used. The sections were digitized using an optical microscope and a scanner. The number of DC-STAMP-positive mononuclear cells and DC-STAMP-positive osteoclasts per ROI were analyzed using Image-Pro Plus. The mean and standard deviation were calculated. 

### 2.4. Cortical Bone Formation Measurement 

CBCT images of the experimental and control groups were obtained using PHT-60FO (VATECH Co., Hwa-sung, South Korea), and the images were reconstructed in a paraxial view using Pacsplus viewer 3.2 (Pacsplus, Orange, CA, USA). As described in our previous study [28], both sides of the paraxial view of the mental foramen were used for measurement. 

#### 2.4.1. Cortical Bone Thickness Analysis 

To analyze cortical bone thickness, the thickness of the cortical bone on the line drawn perpendicular to the mandibular inferior border in the left and right inferior alveolar nerve canal was measured (Figure 2➀).

#### 2.4.2. Cortical Bone Ratio Analysis 

To analyze the cortical bone ratio, the total bone (Figure 2➁) and cancellous bone thickness (Figure 2➂) on the line parallel to the occlusal plane above the mental foramen on both sides were measured. The thickness of the cortical bone was obtained by subtracting the thickness of the cancellous bone from that of the total bone. Then, mandibular cortical bone ratio was calculated as the percentage of thickness of cortical bone from that of total bone. 

### 2.5. Statistical Analysis 

Kolmogorov–Smirnov test was conducted to assess the normality of the distribution, and Mann–Whitney U test was performed to test the statistical hypothesis. SPSS Statistics 27 (IBM, New York, NY, USA) was used for all statistical analyses, and *p*-value < 0.05 was considered statistically significant. 

## 3. Results

### 3.1. Histomorphometric Analysis and Histological Findings

#### 3.1.1. Hematoxylin and Eosin (H&E) Staining

In H&E staining, the number of osteoclasts per ROI was higher in the control group than in the experimental group; however, there was no statistically significant difference. The osteoclast diameter was 21.2 ± 1.8 µm in the experimental group and 23.4 ± 2.7 µm in the control group. There was no significant difference in osteoclast diameter between the two groups. Similarly, the number of osteoclast nuclei was similar between the two groups at 3.7 ± 0.4 in the experimental group and 3.8 ± 0.6 in the control group (Table 2). Oval-shaped osteoclasts with multiple condensed nuclei were observed away from the bone surface. Additionally, a ruffled border, which indicates osteoclast activity, was not observed in these osteoclasts (Figure 3).

#### 3.1.2. Tartrate-Resistant Acid Phosphatase (TRAP) Staining

A positive response to TRAP is indicated by red-colored cells. Although a TRAP-positive response is mainly observed in osteoclasts, non-osteoclastic cells may also show a positive response. Therefore, cells with three or more nuclei were considered osteoclasts. TRAP-positive osteoclasts were observed in both the experimental and control groups. The number of TRAP-positive osteoclasts per ROI was 5.9 ± 4.8 and 12.2 ± 9.01 in the experimental and control groups, respectively. It was significantly less in the control group than in the experimental group (Figure 4).

### 3.2. Immunohistomorphometric Analysis (DC-STAMP Expression)

DC-STAMP-positive cells showed brown cell membranes and cytoplasm and were observed in both experimental and control groups. The number of DC-STAMP-positive cells was 78.1 ± 22.7 in the experimental group and 147.3 ± 57.4 in the control group. It was observed to be significantly less in the experimental group than in the control group (*p* < 0.05) (Figure 5). The number of DC-STAMP-positive osteoclasts was 2.3 ± 1.7 in the experimental group and 3.2 ± 1.9 in the control group. Although the number of DC-STAMP-positive osteoclasts was less in the experimental group, there was no significant difference between the two groups (*p* > 0.05).

### 3.3. Cortical Bone Formation Measurement (Cortical Bone Thickness and Ratio)

The thickness of cortical bone was 4.89 ± 1.26 mm in the experimental group, which was significantly greater than 3.41 ± 0.69 mm in the control group (*p* < 0.05). The ratio of cortical bone was also significantly higher in the experimental group at 46.13 ± 10.4% than in the control group at 35.38% ± 4.87% (*p* < 0.05) (Table 3).

## 4. Discussion

Hyperglycemia reduces osteoclast activity, as shown in many previous studies [30,31,32,33]. In 2008, Wittrant et al. [34] reported for the first time that hyperglycemia reduces the differentiation and activity of osteoclasts. Subsequently, other studies have reported that a high glucose concentration also decreases the differentiation and bone resorption of osteoclasts [35,36]. Xu et al. [37] observed that hyperglycemia reduced the expression of molecules, such as DC-STAMP, that play a key role in osteoclast differentiation and inhibited RANKL-induced osteoclastogenesis. Similarly, Dong et al. [38] also reported that the expression of TRAP decreased at a high glucose concentration. 

In this study, we observed that the expression of DC-STAMP-positive mononuclear cells was significantly lower in the experimental group than in the control group. Osteoclasts are generated by the cell–cell fusion of pre-osteoclasts, which are mononuclear cells [39]. This process is essential for the maturation of osteoclasts and the reconstitution of the cytoskeleton [23]. Previous findings demonstrated that pre-osteoclasts have no bone resorption ability in in vitro culture experiments [23], and mice with a defective fusion of pre-osteoclasts developed osteoporosis [40]. Therefore, the fusion of pre-osteoclasts is an important process for the formation of osteoclasts, and the results of our study suggest that there was a limited fusion of pre-osteoclasts in the experimental group. Moreover, T2D further inhibited osteoclast differentiation in patients with MRONJ. However, the number of DC-STAMP-positive osteoclasts was not significantly different between the two groups and was low in both groups. This may be attributed to the decreased expression of DC-STAMP during the fusion of pre-osteoclasts in the process of their differentiation into multi-nucleated osteoclasts [41].

The number of TRAP-positive osteoclasts was also significantly lower in the experimental group. TRAP is expressed during the differentiation of fused pre-osteoclasts into osteoclasts and during bone resorption by osteoclasts [21]. Our results indicate that the bone resorption activity of osteoclasts in the experimental group was lower than that in the control group. It is thought that T2D reduced the bone resorption activity of osteoclasts in MRONJ patients. 

A histological examination also showed no significant differences in the size of osteoclasts and the number of nuclei between the experimental and control groups. The total number of osteoclasts was lower in the experimental group; however, there was no significant difference between the two groups. Although the number of pre-osteoclasts was smaller in the experimental group, there was no significant difference in the total number of osteoclasts between the two groups. These findings may be attributed to several factors. First, most of the osteoclasts observed in the two groups were separated from the bone surface and were round in shape without a ruffled border. Such osteoclasts with unusual morphology were included in the measurements, which may have affected the results. These altered osteoclasts are related to the administration of BP and are mainly observed in MRONJ patients [42]. A ruffled border is formed when osteoclasts resorb bone from the bone surface. BP interferes with farnesyl pyrophosphate synthase (FPPS) and the mevalonate pathway to separate osteoclasts from the bone surface and prevent the formation of a ruffled border [43,44]. Osteoclasts with an abnormal shape are unable to resorb bone, which may lead to a low number of TRAP-positive osteoclasts compared with the total number of osteoclasts. 

Feng et al. [45] demonstrated that the reduction in multi-nucleated osteoclasts within a hyperglycemic environment leads to the immaturity and dysfunction of osteoclasts, reducing the efficient removal of damaged bone. In addition, the activity of osteoclasts is not limited to bone resorption. Karsdal et al. [46] showed that osteoclasts can be a source of anabolic signals for osteoblasts, and osteogenesis may be initiated by the activity of osteoclasts. This suggests that the reduced number of osteoclasts in diabetes patients may lead to a decrease in bone metabolism. This can further promote bone mineralization and increase bone brittleness and fractures [47]. In fact, in previous studies, patients with T2D showed a high rate of fractures despite there being no decrease in bone density [7,8,48]. 

In this study, we observed that the thickness of cortical bone of the mandibular foramen was significantly increased in the experimental group with T2D as compared with the control group. T2D causes a decrease in osteoclast function, which may have led to secondary bone mineralization and an increased thickness of the cortical bone. However, the thickness of the cortical bone may also be affected by intracortical remodeling, as well as the height, weight, and age of patients [49]. Thus, the thickness does not fully reflect increased bone density. As a result, we additionally investigated the ratio of cortical bone of the upper mental foramen, which was significantly higher in the experimental group than in the control group. To measure the cortical bone ratio, the upper mental foramen was selected as the standard, as MRONJ patients showed an invasion of the alveolar and cortical bone due to lesions. Moreover, the alveolar bone was already resorbed due to the edentulous ridge and aging. Thus, the remaining basal bone had to be used as the standard. In addition, as BP has systemic effects, it was suspected that the ratio of cortical bone of the upper mental foramen would not be significantly different from that of other regions. 

Hyperglycemia is a characteristic symptom in diabetes patients. The effects of hyperglycemia on the differentiation and activity of osteoclasts have been demonstrated in many previous studies [34,35,36,37,38]; however, there is still a debate on the effects of hyperglycemia. Only a few studies have investigated the pattern in clinical patients with both T2D and MRONJ. In particular, to the best of our knowledge, there is no study on the immunological analysis of samples obtained during surgeries. However, previous studies [31,32,33,34,37,38] have shown that hyperglycemia reduces the differentiation and activity of osteoclasts in mice and in vitro experiments; our study also had similar findings in actual patients.

In conclusion, in postmenopausal patients who were administered BP for osteoporosis and developed MRONJ as a consequence, T2D inhibited the differentiation and activity of osteoclasts and increased the thickness and ratio of cortical bone. However, several limitations must be considered in the interpretation of this study’s findings. Only a small number of patients had both T2D and MRONJ, and at the time of surgery, the same amount of bone tissue was not collected from every patient, as this study was retrospective in nature. Through follow-up studies, data on T2D and MRONJ patients need to be accumulated to establish T2D as a factor for the prognosis and treatment of MRONJ. 

## 5. Conclusions

Results from our study showed that the osteoclasts of postmenopausal patients with MRONJ and T2D did not show significant quantitative or morphological changes compared with postmenopausal patients with MRONJ who did not have T2D. However, in patients with both MRONJ and T2D, osteoclast differentiation and activity were reduced. Additionally, the thickness and ratio of cortical bone were increased. 

Although this study could not demonstrate that T2D is a risk factor for MRONJ in patients with osteoporosis, it is suspected that postmenopausal patients with osteoporosis who have T2D and BP-induced MRONJ may have dysfunctional osteoclasts and decreased bone quality due to increased cortical bone formation, which may increase the risk of delayed healing and fracture. Therefore, additional attention and observation with a view toward a cure for postmenopausal MRONJ patients with T2D are necessary. 

## Figures and Tables

**Figure 1 jcm-11-02377-f001:**
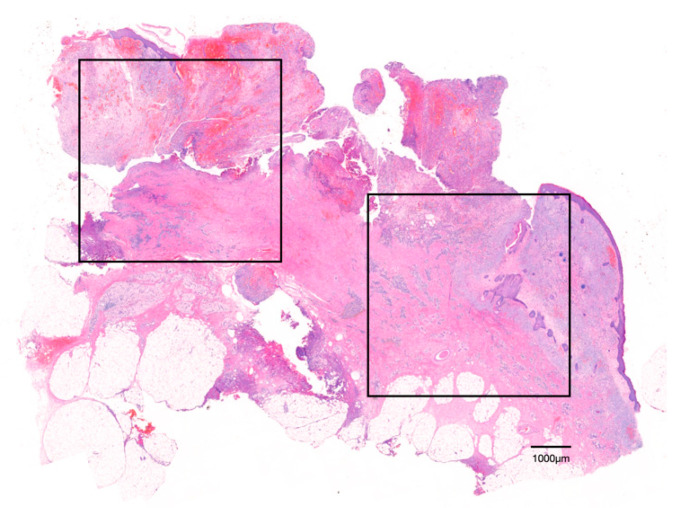
Region of interest (ROI) is indicated. Two visual fields (rectangles) were defined and measured within a scanned section (H&E staining).

**Figure 2 jcm-11-02377-f002:**
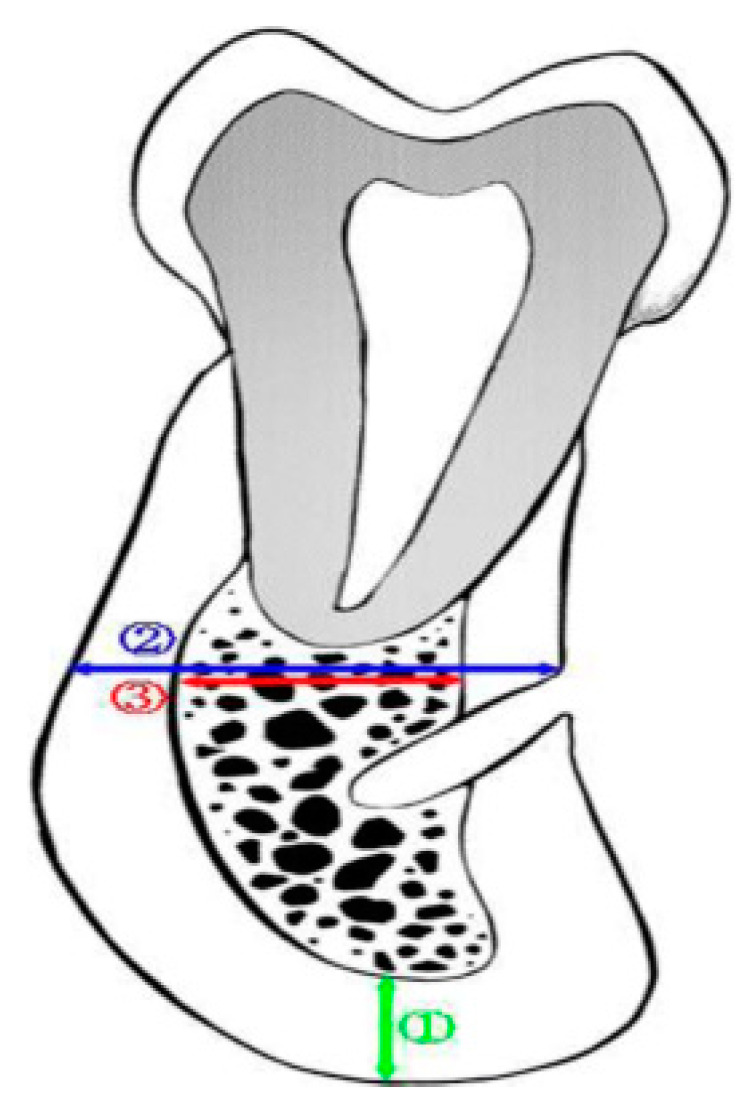
Paraxial view of the mental foramen in CBCT. **➀** Thickness of the cortical bone in mandibular inferior border. **➁** Thickness of the total bone on the line parallel to the occlusal plane. **➂** Thickness of the cancellous bone thickness on the line parallel to the occlusal plane.

**Figure 3 jcm-11-02377-f003:**
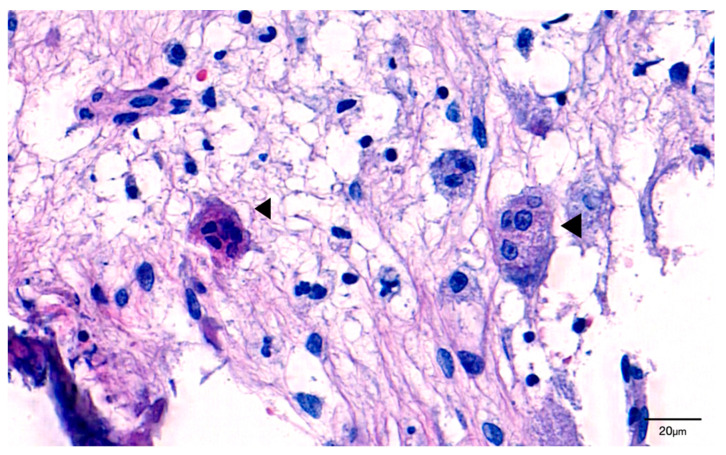
Giant hyper-nucleated osteoclast (arrowhead) within a MRONJ histological section (H&E staining). Note the pyknotic nuclei, round shape, detachment from the bone surface, and lack of ruffled borders.

**Figure 4 jcm-11-02377-f004:**
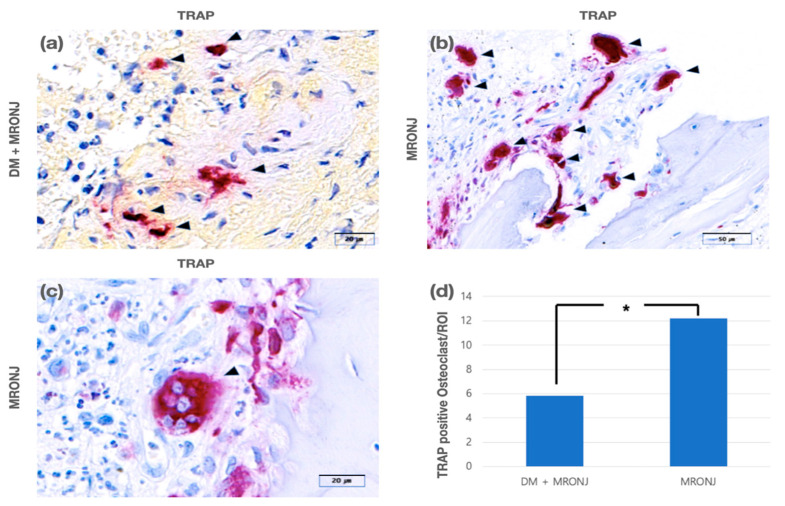
(**a**) TRAP-positive osteoclasts (arrowhead) in the DM + MRONJ group (TRAP staining). (**b**) TRAP-positive osteoclasts (arrowhead) in the MRONJ group (TRAP staining). (**c**) TRAP-positive giant osteoclasts in the MRONJ group (TRAP staining). (**d**) There were significantly fewer TRAP-positive osteoclasts in the DM + MRONJ group than in the MRONJ group (* *p* < 0.05).

**Figure 5 jcm-11-02377-f005:**
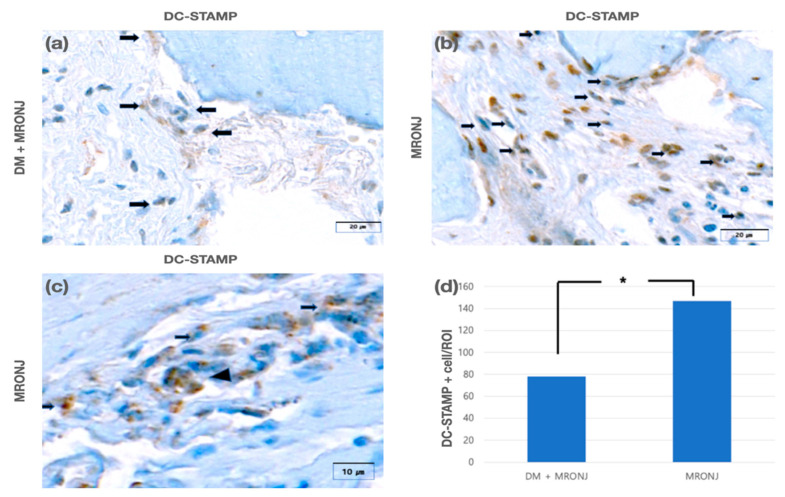
(**a**) DC-STAMP-positive cells (arrow) in the DM + MRONJ group (anti-DC-STAMP staining). (**b**) DC-STAMP-positive cells (arrow) in the MRONJ group (anti-DC-STAMP staining). (**c**) DC-STAMP-positive osteoclasts (arrowhead) in the MRONJ group (anti-DC-STAMP staining). (**d**) There were significantly fewer DC-STAMP-positive mononuclear cells in the DM + MRONJ group than in the MRONJ group (* *p* < 0.05).

**Table 1 jcm-11-02377-t001:** Baseline characteristics of patients (*n* = 20).

	Experimental Group (T2D + MRONJ)	Control Group (MRONJ)	*p* Value ^1^
Number of patients	10	10	-
Gender	F	F	-
Age	76.9 ± 4.1	77.0 ± 6.5	>0.05
T2D duration, year	11.7 ± 5.9	-	-
HbA1c, %	7.3 ± 0.7	-	-
BP duration, year	5.4 ± 1.8	6.7 ± 2.8	>0.05

T2D, type 2 diabetes; BP, bisphosphonate; F, female. ^1^
*p* values were obtained via the Mann–Whitney U test.

**Table 2 jcm-11-02377-t002:** Histomorphometric and quantitative analysis of patients (H&E staining).

	Experimental Group (T2D + MRONJ)	Control Group (MRONJ)	*p* Value
Diameter of osteoclasts (μm)	21.2 ± 1.8	23.4 ± 2.7	>0.05
Nuclearity of osteoclasts (nuclei/osteoclast)	3.7 ± 0.4	3.8 ± 0.6	>0.05
Osteoclasts per ROI	42.9 ± 13.2	45.7 ± 18.4	>0.05

**Table 3 jcm-11-02377-t003:** Cortical bone thickness (mm) and ratio (%) in patients.

	Experimental Group (T2D + MRONJ)	Control Group (MRONJ)	*p* Value
Cortical bone thickness (mm)	4.89 ± 1.26	3.41 ± 0.69	<0.05
Cortical bone ratio (%)	46.13 ± 10.42	35.38 ± 4.87	<0.05

## Data Availability

The data presented in this study are available on request from the corresponding author.

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
