# Peer review of "Effects of Type 2 Diabetes Mellitus on Osteoclast Differentiation, Activity, and Cortical Bone Formation in POSTmenopausal MRONJ Patients"

_jcm, 2022, doi:10.3390/jcm11092377_

Round 1

Reviewer 1 Report

Dear authors,

It was a pleasure to read the scientific article entitled ” Effects of type 2 diabetes mellitus on osteoclast differentiation, activity, and cortical bone formation in postmenopausal MRONJ patients”. Congratulations of your work!

From my point of view the study is very interesting, well conducted, the materials and methods are clear and represented in a scientific manner. 

On the other hand, I am concerned only on an aspect that must be clarified: line 177- ”p-value >.05 was considered statistically significant”. Based on my knowledge, a p-value of 0.05 or lower is generally considered statistically significant. Please check this aspect. 

Author Response

Response 1 :

Dear Reviewer

It is a great honor for me to revise and resubmit our manuscript JCM.

The changes are highlighted in the manuscript using red colored text.

We sincerely apologize for the typing mistake. 

We look forward to your response.

Sincerely yours,

Jae-Hoon Lee, DDS, PhD

Professor

Department of Oral and Maxillofacial Surgery.

College of Dentistry, Dankook University

119 Dandae-ro, Dongnam-gu, Cheonan, South Korea

TEL: +82-41-550-1996, FAX: +82-41-551-8988

E-mail: lee201@dankook.ac.kr

Reviewer 2 Report

This article addresses the important problem of the jaw bone osteonecrosis in postmenopausal patients treated with bisphosphonates for osteoporosis and osteopenia with or without type 2 diabetes. The aim of the work is clearly defined, and the research methods used are sufficiently described. The presentation of the results does not raise any doubts for a reader. The discussion is clear and based on the most recent articles regarding the problem. The authors proved in the article that they are well prepared for the completion of this type of research projects.

Minor remarks of the reviewer that may improve the substantive value of the article:

  • being a practicing maxillofacial surgeon, I lack practical guidelines for doctors dealing with patients with osteonecrosis of the jaws treated with bisphosphonates and suffering from diabetes, even if at the present stage of knowledge these remarks would be questionable
  • defining the aim of the study, the authors should add the information about the area of ​​the bone in which the thickness of the cortical bone and the ratio of the cortical bone was measured (line 71)

Author Response

Point 1 :Being a practicing maxillofacial surgeon, I lack practical guidelines for doctors dealing with patients with osteonecrosis of the jaws treated with bisphosphonates and suffering from diabetes, even if at the present stage of knowledge these remarks would be questionable.

Response 1: Thank you very much for your good point.

Unfortunately, it is true that our study does not offer a practical treatment or guideline for patients with both MRONJ and T2D. 

Instead, we focused on the advantages of the oral and maxillofacial surgery area, which can obtain bone specimens and CBCT at the same time.

Based on this, we found  out that osteoclast differentiation and activity were reduced and the thickness and ratio of cortical bone were increased in postmenopausal patients with MRONJ and T2D.

We conclude that these findings may increase the risk of delayed healing and fracture in patients with MRONJ and T2D. 

We believe that the significance of this study is to establish this facts in actual patients using samples obtained during surgery and CBCT.

Point 2: Defining the aim of the study, the authors should add the information about the area of ​​the bone in which the thickness of the cortical bone and the ratio of the cortical bone was measured (line 71)

Response 2: We defined the area of the bone as near the mandibular mental foramen. The changes are highlighted in the manuscript using red colored text.